# Integration of Building Information Modeling (BIM) and Virtual Design and Construction (VDC) with Stick-Built Construction to Implement Digital Construction: A Canadian General Contractor's Perspective

**Mohamed Adel [1], Zhuo Cheng [2] and Zhen Lei [2,*]**

1   Bird Construction, Halifax, NS B4A 2Z2, Canada
2   Off-Site Construction Research Centre (OCRC), Department of Civil Engineering, University of New Brunswick, Fredericton, NB E3B 5A3, Canada
*   Correspondence: zhen.lei@unb.ca

**Abstract:** Building information modeling (BIM) and virtual design and construction (VDC) are useful management processes and methodologies to enhance project communication and coordination. Over the past few decades, researchers and practitioners have made efforts to promote the adoption of BIM and VDC in the construction industry. However, currently, the promotion and adoption of BIM and VDC are relatively slow in North America. This paper focuses on developing an approach to drive the adoption of the technologies through cooperation among project stakeholders and proposing a method of collaboration through a case study. In this paper, the authors surveyed and interviewed a major Canadian general contractor on its implementation of BIM and VDC in construction projects. The was to benchmark the status of BIM and VDC applications in the Atlantic region of Canada from a general contractor's perspective. Both surveys and interviews were conducted with various project participants throughout the organization. Based on the results, a "Digital Construction Framework for the Future" is proposed to increase the adoption of BIM and VDC. This research can provide a general understanding of BIM and VDC adoption in this particular general contractor and areas of improvement in transitioning to a more digital working construction environment.

**Keywords:** building information modeling (BIM); virtual design and construction (VDC); construction industry benchmarking; digital construction



## 1. Introduction

The architecture, engineering, and construction (AEC) industry, as a traditional industry, has played an important role in our society for a long time. More and more comprehensive and large-scale constructions thrive in cities. Most individual AEC projects are divided into small packages (e.g., a work breakdown structure), and many parties with different functions are involved in various tasks. However, when facing complex projects, it is difficult to reach comprehensive management: the sequence of tasks in construction involves space, resources, time, procurement, dimension constraints, and other issues in the project process [1]. Additionally, participants of the same project tend to be relatively fragmented, with a lack of communication, collaboration, and integration between parties [2]. This fragmentation of construction industry participants causes major problems [3], such as repetitive work, the waste of resources and labor, cost increase, construction schedule delay, and even safety risks. Consequently, inadequate planning and information loss during project delivery are the main problems that reduce productivity and quality and increase cost, time, and the potential risks. Solving these problems has been identified as the main goal that the AEC industry currently wants to achieve. Given this, the concept of "digital construction" was proposed and is gradually being implemented to complete the traditional construction industry [4].

Digital construction technology is the use of a computer-generated model to simulate the planning, design, construction, and operation of a facility across the lifecycle of a project, a technology that allows users to create a visual simulation of a project with a digital prototype of a building prior to construction [5]. In the facilitation of digital construction technology, building information modeling (BIM) and virtual design and construction (VDC) have emerged and have been developed to meet the needs of the industry. Since first proposed in the early 2000s, BIM and VDC are the main technologies currently used to achieve digital construction implementation. The deployment of BIM/VDC in construction projects can make the industry more efficient and flexible in project management. However, the implementation of BIM and VDC still faces problems, such as adoption. Achieving an increase in the application of BIM and VDC technologies in actual construction projects and drive their development is a pursuit that needs attention. Many case studies have shown that the application of BIM/VDC technology could bring not only more profit margins to the contractors, but also provide non-financial advantages to society (e.g., nonrenewable resource conservation, pollution reduction, and so on). Thus, the construction industry should not be solely responsible for the introduction, acceptance, and promotion of these new technologies; many other parties need to provide support (e.g., government, etc.). With more support from different fields, adoption of the technology can undergo faster and better development. Research is required on which parties should participate in promoting the use of these technologies and the methods by which they collaborate. Therefore, the main topic of this paper is to clarify the responsibilities of each party and form a mutually supportive and cooperative relationship to drive the implementation and development of BIM and VDC technologies through a case study.

This paper will commence with a review of current BIM and VDC implementation trends, benefits, and challenges in the AEC industry. Collaborating with Bird Construction, one of the largest general contractors in North America, a case study was conducted. The interview was conducted with the BIM/VDC manager who oversaw the digital construction implementation of this project, alongside a follow-up questionnaire with the participants of the same project. Based on the opinions obtained from the industry practitioners, this research mainly focuses on analyzing the status quo of BIM and VDC and how to promote the adoption of these technologies in the future. The interview questions were divided into three parts (technology, process, and culture), and the questionnaire questions were about the usage of BIM/VDC technology and skill training. The interview and questionnaire results were summarized and categorized into three individual parties: industry, institution and government, and education. These three parties are inseparably interconnected and supplementary to each other. A new framework, the "Digital Construction Framework for the Future," was proposed to conclude and describe the relationship between and duties of these three parties, aiming to improve the adoption of digital technologies in future directions. Given that the research is conducted based on a single general contractor in Canada, it does not reflect the entire AEC industry. This is a limitation of this research. However, by its nature, the research provides general guidance for the application of BIM/VDC in the AEC industry through the viewpoint of practitioners. As such, the results have both academic and practical values for those who plan to adopt BIM/VDC in their organizations. To overcome this limitation, the authors will expand the survey areas in future work (e.g., covering other companies and different provinces in Canada).

## 2. Methodology

In order to study the current state of the usage BIM and VDC technologies in the construction industry, a mixed-method approach based on both qualitative (interviews) and quantitative (questionnaires) methods was designed. As shown in Figure 1, there were two phases designed to construct the final framework. Phase 1 included a literature review as a preliminary investigation, mainly conducted by collecting and researching the application of BIM and VDC and workflow literature to form an initial understanding of their history and status. The specific content included the definitions of BIM and VDC,

development, the economic and non-economic benefits that they could provide, and the challenges of further promotion. After this investigation, a general understanding of BIM and VDC adoption could be obtained, and a general logic for the creation of this research framework was formed. After Phase 1, the research methods were designed, focusing on a mixed community project in Halifax, NS, Canada. The research group cooperated with Bird Construction to conduct Phase 2, a case study including an interview and an online questionnaire. The goals were to understand the entire process and the roles that technologies played in real construction cases, obtain opinions from a Canadian general contractor in the Atlantic region's perspective, and conduct a follow-up analysis.

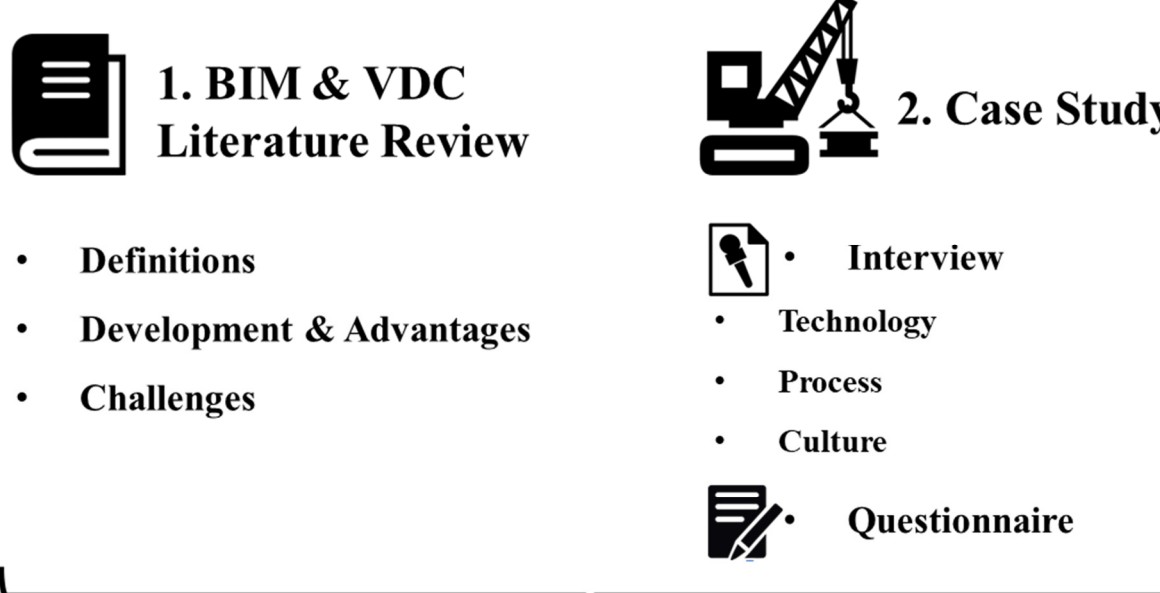

**Figure 1.** Methodology of researching and proposing a framework for promoting development.

Bird Construction, the studied company, is one of the leading general contractors in construction in North America. The company develops and builds residential and commercial properties, industrial facilities, public transportation, civil engineering structures, and other types of infrastructure. Their scope of work includes general contracting, construction management, design–build, public–private partnerships, engineering, procurement and construction, pre-construction, BIM, sustainable design and construction, and so on. In this paper, the case study project was constructed by Bird Construction. This paper focuses on the adoption of BIM and VDC technologies in the construction field and its future development. As such, the main participants of this research were the BIM/VDC practitioners. The interview subject was a BIM/VDC manager who oversees digital construction implementation throughout the case study project, and the interview was aimed at analyzing the implementation and utility of BIM and VDC in three parts: technology, process, and culture. In the technology part, the BIM and VDC software currently in use in the industry were summarized and categorized by type, function, phase, and owner/user. In the process part, the procedure of the use and function of these technologies throughout the whole project was explained by the BIM/VDC manager. The role of clients, project stakeholders, and other different departments was also included in this part. These two parts were used to discuss the industry party's duty in the digital construction framework. Culture, the third part, was about the changes brought by the application of BIM and VDC

to the industry along with the means by which to promote their adoption. Within the third part, the need for support from institutions and the government was mentioned. The research questionnaire was anonymous, directed at the team members who participated in this project to obtain their opinions on the implementation of BIM/VDC technology on real projects and job-related skills training. In the training questions, the concern of the education party in the framework was raised.

## 3. Literature Review

Building information modeling (BIM) was introduced in pilot projects in the early 2000s [6] and it gained widespread use after Jerry Laiserin (a construction industry analyst) argued that it should be a standard industry term in 2002. After that, the term "BIM" attracted Autodesk, who started heavily promoting it alongside their products [7]. BIM is generally understood as an overarching term to describe a variety of activities in object-oriented computer-aided design (CAD), which supports the representation of building elements in terms of their 3D geometric and non-geometric (functional) attributes and relationships [8]. Additionally, based on traditional 2D drafting and 3D modeling, it allows for the application of fourth and fifth dimensions related to time and cost estimation [9]. It is defined as the process of generating, storing, managing, exchanging, and sharing building information in an interoperable and reusable way [10]. During the process, it collects fragmented information, links them with the as-built model, and forms an accessible database for everyone involved on this project. Additionally, the BIM model is usually updated and shared between the project participants along with the different phases of the project from planning to commissioning [7] to reach more efficient exchanging and updating of project data via communication and information management [11]. Thus, BIM refers to a set of technologies and solutions aiming at enhancing inter-organizational collaboration in the construction industry, which will improve productivity whilst improving the design, construction, and maintenance practices [8].

Virtual design and construction (VDC) is an important part of BIM design. It provides virtualization schedule animation in time and/or task order and presents the building under construction from start to finish. The BIM-based model components usually contain production rate information (for all the associated tasks) that will permit a line of balance schedule analysis; this approach allows the fine-tuning of tasks based on their location in the project and production rates, and helps to eliminate start and stop cycles within tasks [12]. Therefore, after VDC technology virtualizes the schedules, project planners can compare various schedules easily and they can quickly recognize whether the project is on track or not [13]. Additionally, it also assists in improving the identification of schedule consequences by providing a graphical representation and the preparation of an opportunity to distinguish unexpected problems and incompatibility conflicts. This approach facilitates the scheduling process and helps to distinguish potential time–space conflicts [14]. Due to these advantages, VDC improves the engagement of project stakeholders, helps to communicate project procedures to the owners, gives subcontractors and tradespeople a better understanding of their work scope and schedule and helps field personnel verify that the project is on track. It is considered to be the main connection tool between the architecture part of the project and planning scheduling in the construction part of the project. To summarize, the adoption of BIM and VDC technologies has been at the core of digital construction management. These digital technologies offer an overall plan throughout the whole project up front which will be used throughout the entire project life cycle. Major benefits consist of design consistency and visualization, cost estimations, clash detection, implementation of lean construction, and improved stakeholder collaboration [15].

Since BIM was first introduced in pilot projects in the early 2000s [6], the implementation of BIM and VDC in the AEC industry has gained momentum globally. Many cases support the fact that the adoption of BIM/VDC technology reduces waste (e.g., material, resources, labor, and cost), reduces risk, shortens the construction period, and increases project quality. These technologies increase clients' satisfaction through the visualization

model and clear expectations, enhance team collaboration in the delivery of better outcomes, and improve data sharing, information control, and the delivery of green buildings [7]. GBP revealed that BIM helped improve the efficiency of the project execution by 37%. It reduced risks on site by 34%, as it allowed for the detection of any conflicts between the different models (e.g., architectural, mechanical electrical and plumbing (MEP) models, and structural models); promoted prefabrication and the manufacturers' integration by 26%; reduced waste by 24%; improved transparency in communication by 21%; and offered the potential for automation to industrialize construction [16]. There are also many real cases demonstrating that BIM and VDC technologies are beneficial for both clients and project stakeholders. The Aquarium Hilton Garden Inn was a project comprised of a mixed-use hotel, retail shops, and a parking deck. The architectural, structural, and MEP BIM models were designed upfront and used throughout the project lifecycle. During the design development phase, 55 clashes were identified, which resulted in a cost avoidance of USD 124,500. During the construction development phase, more than 590 clashes were detected before actual construction began. The net cost saving was roughly considered to be USD 200,392, while the development cost of the original BIM model was USD 90,000. It exceeded the expectations of the owner and other project team members, so this project was seen as a successful implementation case of BIM technology. Savannah State University in Georgia was another successful case illustrating the use of BIM during the project planning phase. In this case, BIM technology showed its advantages: it improved performance outcomes by comparing different design alternatives, reduced information errors and omissions, reduced rework and safety risks, and facilitated precise scheduling [17]. Prior to the construction, there were three different design options alongside their value analysis for clients and project stakeholders to select the most economical and workable building layout. Additionally, VDC technology was applied to form walk-in virtual models to help decide the best option that fits their requirements. Finally, the cost benefit was estimated to be $1,995,000, approximately 400 times that of the BIM/VDC cost (USD 5000) [18]. In addition, Azhar [18] did a return on investment (ROI) analysis based on 10 projects built from 2005 to 2007. The average BIM ROI for projects under the same study was 634%, which clearly depicted its potential economic benefits. Except for the construction part, Akcamete et al. [19] indicated that the cost of operation and maintenance of the building, the role of the facilities managers, equated to 60% of the overall costs of the project, while only 15% of the overall costs were spent on BIM design. They further suggested that great financial gains could be achieved by targeting this aspect of a project: if BIM and VDC technologies could be effectively applied to reduce the potential maintenance in the future or could be used directly for maintenance, the project costs could be significantly saved in the long term. In conclusion, the economic and quality benefits that BIM/VDC technology could bring are widely acknowledged and increasingly well understood. In addition to the reported savings, there were also unknown costs avoided through collaboration, visualization, understanding, and early identification of conflicts [20]. Thus, the economic benefits BIM/VDC could bring to the owners and project stakeholders were significant.

Besides the economic benefits that digital design can bring, BIM and VDC technologies are also seen to have the ability of safety control. In the past two decades, more than 26,000 U.S. construction workers have died at work, which equates to approximately five construction worker deaths every working day [21]. Safe construction requires careful planning throughout the project lifecycle, from design to construction planning, to construction execution and extending, and to operations and maintenance. Since good safety practices and records create a positive, hazard-free, and productive work environment, planning for safety at the beginning of a project is not only the first but also the fundamental step for safety management [22]. The growing implementation of BIM in the AEC industry is changing the way that safety planning is approached. Kang et al. [23] proposed building a 5D CAD-based risk visualization system for visualizing the degree of construction risk. Furthermore, Zhang et al. [17] designed an automated rule-based checking system for safety planning and simulation by applying algorithms to BIM. By integrating the new technology,

BIM and VDC showed feasibility in automated hazard identification and correction during the construction planning phase. In addition, as previously mentioned, the detection of spatial conflicts or congestions of construction operations was one issue addressed using 4D visualizations [24]. The VDC 4D modeling with safety in the scheduling process could monitor the safety program on the site. It performed defect identification of the program on time before an accident and injury to decrease the rate of accidents on the site [2]. To conclude, BIM and VDC have enabled virtual safety controls to be used to identify safety hazards. They have the potential to simulate various stages of the construction process to help engineers, architects, and contractors to detect, visualize, and resolve risks prior to the problematic conditions arising on the project [17].

However, the adoption of digital technologies in the AEC industry, including BIM and VDC, was slower than anticipated [18]. There are still many existing challenges hindering the widespread implementation of BIM/VDC technology. Saka and Chan [3] concluded that the major barriers of adopting BIM technology in small and medium-sized enterprises (SMEs) in the UK were the lack of government support, lack of BIM knowledge, lack of stakeholders' awareness, and the high cost of BIM implementation. To be more specific, first, the BIM model can be part of an extranet [25], which can lead to legal issues. There was a need to deal with the legal issues through the construction contract in order to reduce this significant risk. Second, the cost of implementing BIM in terms of resources and training had been seen to be a substantial barrier within the construction industry [26]. If the company did not make a rigorous plan before BIM implementation, financial issues associated with the initial investment required, technological issues related to BIM tools, and cultural and management barriers linked to human resources are likely be faced [27]. Although BIM will ultimately be driven by clients [20], demonstrating the benefits of these new technologies and convincing customers to accept and even be willing to pay for them is a big problem. Thus, "who should pay for the development of digital construction" was a contentious question. Hore et al. [28] suggested that if adoption became a requirement, then government should take the main responsibility to facilitate implementation. Research revealed and confirmed that the most critical factor for successful BIM implementation was national leadership and coordination to maximize efficiency and avoid the many problems created by piecemeal and disjointed approaches. This leadership should primarily be driven by government entities, but it also needs the support of and collaboration with major industry parties such as major private sector clients, contractors, and industry/professional associations [29]. The European Union Public Procurement Directive (EUPPD) was signed by the European Parliament for the 28 European Union member countries to encourage, specify, or mandate the use of BIM for their publicly funded projects [29]. With support from the government, as the AEC industry stakeholders, the industry party should play the main role in improving BIM performance. Furthermore, the visual nature of BIM/VDC allows the project team to be involved in the virtual model for better cost and schedule estimation, risk mitigation, and quality control. Besides support from the government, optimizing BIM performance would undoubtedly be a challenge for industry practitioners which can increase the direct and indirect net income made by digital construction implementation to improve its competitiveness. Third, Eadie et al. [20] conducted a survey regarding the "reasons for not using BIM in projects". The top two reasons were "Lack of expertise within the project team" and "Lack expertise within the organizations". This conclusion indicated that the issue education was another main barrier that retards digital technology development in the AEC industry. It was also worth mentioning that 82.61% of participants considered the implementation of BIM/VDC technology to have been beneficial on projects, which indicated that their advantages were widely acknowledged. Courses and tests are needed for applied knowledge and practical experience. Either companies, schools, or both need to find a way to lessen the learning curve of BIM trainees and create a universal standard to evaluate practitioners' skills.

To summarize, BIM and VDC technologies are two significant digital construction management approaches which were proposed and designed to address problems caused

by the fragmentation of the current construction industry. After several years of development, they have been well developed and their beneficial advantages for both clients and project stakeholders are indicated by many real construction project cases. Their advantages include a direct reduction in economic cost and many other things (e.g., construction schedule reduction, cost savings, construction sustainability, future maintenance, and risk mitigation). In spite of BIM and VDC technologies being obviously beneficial for the AEC industry, their usage is not as frequent as people predicted and there are still many barriers hindering their development.

## 4. Case Study

The University of New Brunswick (UNB) research team has conducted an interview as well as a questionnaire with Bird Construction, one of the largest general contractors in North America. The focus of the interview and questionnaire is the Atlantic region of Canada (i.e., three Maritime Provinces—New Brunswick, Nova Scotia, and Prince Edward Island, Canada). The research interview and questionnaire participants are from the Atlantic region of Canada and the construction project used in the case study is located in the Atlantic region. It should be noted that this research aims to capture the adoption level of BIM and VDC technologies in the Atlantic region, and the results could be limited to a certain extent due to the group size of the interview and questionnaire. However, the results can be used to set a benchmark of general understanding of BIM and VDC adoption in the Atlantic region of Canada. Additionally, the names of the project and participants are excluded in this paper anonymously.

The case study project is located in Halifax, Nova Scotia, Canada, a mixed community, with two phases of construction. Phase 1 work consists of two levels of underground parking that encompass the entire site, a 33-story tower, a 12-story tower, an 8-story tower and three other buildings capped at the podium level. These towers will contain approximately 100,000 sq. ft. of commercial space and 400 residential units. Phase 2 work will see the three capped buildings finished along with the green spaces and landscaping. Upon completion, the project contains around 74,676 sq. ft of office space, 460,000 sq. ft of retail space (including eight live/work studios), 500 residential units, and a parkade (containing 600 underground and 50 surface parking spaces). The project duration is 44 months, with more than 100 million dollars in construction value. The project delivery method is construction management, in which Bird Construction served as the construction manager of the project. See Figure 2 for the rendering of the project case.

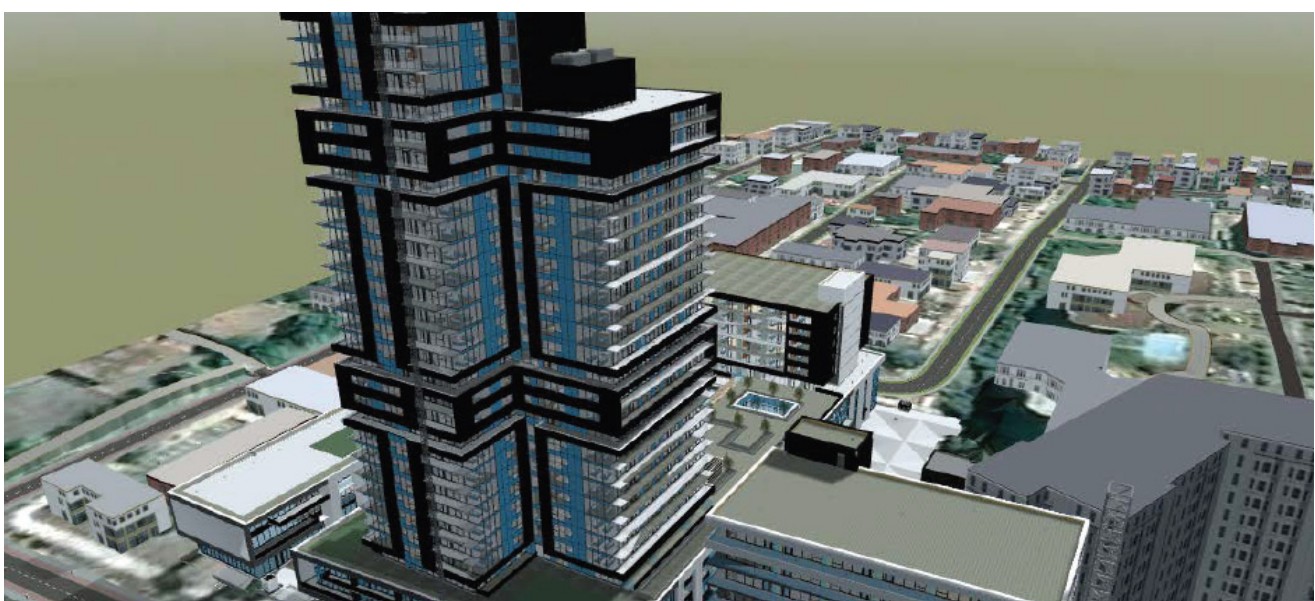

**Figure 2.** Project case rendering.

*4.1. Interview*

4.1.1. Interview Structure

In the interview, the UNB research team interviewed the BIM/VDC manager who oversaw the implementation of digital technologies throughout the project. The interview was structured to target three areas: (1) technology, with the intention of capturing the technological applications used in this project and their functionalities; (2) process, with the intention of capturing the process of implementing BIM and VDC technologies; and (3) culture, with the intention of capturing the culture shift of the project participants throughout the implementation of the technology. The detailed questions are listed as below:

Technology-related questions:

1.     What BIM/VDC technologies were used throughout the project?

Process-related questions:

1.     What are the thoughts and processes of selecting technologies?
2.     What are the measures of the technology's success/failure?
3.     What is the BIM/VDC team composition? And project management mechanism?

Culture-related questions:

1.     What is the culture of BIM/VDC before the project?
2.     What is the culture of changes after the project?
3.     What are the barriers of culture, if any?
4.     What are the enablers of culture changes, if any? For future improvements?

4.1.2. Results

Through the interview, the research team captured the technologies used in the project, as shown in Table 1.

For the process of implementing the BIM and VDC technologies, the general contractor has initially invested in and initiated the BIM implementation for this project, which has been set up as a pilot project for clients to understand the value of BIM. The aftermath of the successful implementation of BIM in this project has led to an increase in the demand by the clients for the upcoming projects. There has been a culture shift from the general contractor initiating BIM to the clients demanding BIM implementation in projects. From the general contractor's perspective, the initial investment in implementing BIM and VDC is directly related to project risk mitigation, coordination between various project participants, and less contingency of the budgeted cost. However, the long-term goal of BIM and VDC implementation in the project is to develop a digitalization-centric culture within the organization. With the implementation of BIM and VDC, the general contractor started to establish key performance indicators to track cost and time savings on the project, with consideration of the impacts on coordination, schedule enhancement, rework reduction, quality improvement, information availability, and estimation accuracy.

In the process of the implementation of BIM and VDC, the general contractor led the way, with the senior VDC coordinator to maintain the modeling and coordinate between the model authoring and the end user. The senior VDC coordinator was also responsible for facilitating various digital coordination and construction environment that the project team was able to use for actual construction. The senior VDC coordinator was considered the "champion" in the BIM and VDC implementation process. In addition, the project coordinators were in support roles to the senior VDC coordinator in the BIM and VDC implementation process, performing tasks including splitting models to reflect actual phasing on site and linking the model to the project schedule, conducting reality capture through 3D scanning, and managing RFIs and document control through a digital platform, i.e., Revizto. The project manager used the model for planning purposes and for confirming quantities with the suppliers/subtrades. The estimator primarily used the model for quantifying project elements and tendering based on quantifications. On the construction sites, the superintendent was also involved with reporting progress through the mobile

device, e.g., Assemble, and used the modeling for visualization purposes. The process of implementing BIM and VDC replied primarily on the initiative of the general contractor, as the engineering team did not have contractual obligations to produce a VDC-oriented model (e.g., a model with a high level of details (LOD)). The suppliers and subtrades did not have obligation in supporting the BIM and VDC implementation; however, they provided 3D models when requested by the general contractor. Some of the suppliers already adapted to some VDC applications in their business (i.e., providing models is business as usual for them). The general contractor s great potential by involving suppliers and sub-trades in the BIM and VDC implementation.

**Table 1.** Technological applications in project case study.

| Type | Technology | Function | Phase | Owner/Users |
|---|---|---|---|---|
| Authoring | Revit Civil 3D | Project modeling | Design and construction | Owner: engineering firm Users: the rest of stakeholders |
| Model hosting | BIM360 | Design collaboration and file sharing | Design and construction | Owner: engineering firm and GC Users: all the stakeholders |
| Model context | Autodesk Infraworks | Importing the project context (e.g., surrounding, layout, etc.)—for visualization and logistics planning | Construction | Owner: GC Users: all the stakeholders |
| Virtual coordination project environment | Revizto | Visualize, coordinate, communicate and track issues; review drawings, manage RFIs and project deficiencies; and change management | Construction | Owner: GC Users: all the stakeholders |
| Virtual construction project environment | Fuzor | Master schedule visualization (4D), static and dynamic logistics, safety orientation, constructability analysis | Construction | Owner: GC Users: all the stakeholders |
| Data classification and management | Assemble | Quantity management, model classification into different divisions (master format), progress tracking, change management | Construction | Owner: GC Users: all the stakeholders |
| Data analytics | Power BI | Dashboard for progress tracking, workload monitoring and projection | Construction | Owner: GC Users: all the stakeholders |
| Reality capture | RTC360 | As-built conditions and quality control | Construction | Owner: GC Users: all the stakeholders |

From the cultural perspective, BIM and VDC implementation shifted the way that project participants thought about project management processes. At the beginning of the project, there was no digital construction culture and the general contractor had to make a decision to invest and lead the way. The barriers of implementing BIM and VDC often came from the status quo and the level of difficulty/ease of use of technology. Through the implementation, a proper workflow is required to ensure the participants can follow the implementation easily and efficiently. Through this project, four areas were identified as enablers for future BIM and VDC implementation in similar construction projects: (1) project life cycle consideration of BIM and VDC, i.e., the implementation should not be limited to the construction phase; (2) more involvement among all project participants, which requires a more integrated project delivery process, e.g., design–build and integrated project delivery (IPD); and (3) government mandates, which can provide incentives for the industry to adopt technologies and innovations; and (4) company-level roadmaps and

initiatives set by the companies to manage and implement BIM and VDC through the entire project management process.

*4.2. Questionaire*

A web-based research questionnaire titled "BIM and VDC Technology Adoption & Implementation Questionnaire" was conducted as supplementary material for this case study. The questionnaire was structured in four parts: (1) background, i.e., participants' current position and the field they were working in; (2) technology, i.e., software used in their daily work; (3) pre-work training; and (4) impacts. The detailed questions are listed below:

Background-related questions:

1. What is your current position?
2. Which field are you mainly working in?

Technology-related questions:

1. What BIM and VDC technologies are involved during your working process?
2. In which project phase do you use it/them?
3. What is the frequency you use it/them during your working process?

Pre-work-training-related questions:

1. Did you receive BIM and VDC training before you take on the current position? How long?
2. Do you feel you need more BIM and VDC training to become proficient in your current position?

Impact-related questions:

1. Compared with traditional project management, what do you think is the impact of digital construction management with BIM and VDC technologies?

In the background-related questions, there were seven participants. They came from Bird Construction and participated in the case study project (one construction manager/project manager, one estimator, one senior construction superintendent, two project coordinators, and two site superintendents). Four construction working fields (estimation, quality control and management, project control, and site management) were involved.

In the technology-related questions, the technologies mentioned by the BIM/VDC manager in the interview were used as the options. As shown in Figure 3, Fuzor and Assemble were the most popular BIM/VDC technologies, which were both chosen by six participants, followed by Revizto (chosen by five participants). Assemble and Revizto are the visualizing coordinating BIM technologies. They are used to offer general contractors and project stakeholders a sharable and real-time interactive platform for coordinating, communicating, modifying, progress tracking, and so on. Fuzor is a VDC technology used for 4D master schedule visualization, logistics analysis, and constructability analysis. Power BI and RTC 360 are two quality control BIM technologies ranking third and fourth in this research questionnaire. Revit is currently one of the most commonly used and popular BIM software. It is often seen as the base of BIM design and many technologies mentioned in this report are closely related to it. However, in this research questionnaire, it ranked fourth. It may be due to the small size of the research group and the non-diversity of participants' working fields since they were mainly from the virtual-digital design team. This result may indicate that the BIM and VDC technologies are continuing to develop, and more and more new technologies have been created. They are designed based on some traditional BIM software, such as Revit, combining more specific and practical functions to meet the specific requirement. Naviswork, Civil 3D, Infraworks, BIM 360, and Auto CAD were not chosen in this questionnaire, which may also be a result of the limitation of this research group. Therefore, a larger research questionnaire is required to support this conclusion. Figure 4 is the summary of project phases when technologies were used. It shows that BIM and VDC technologies are mostly used in the planning and construction phases, which concurs with the interview and Table 1.

What BIM and VDC technologies are involved during your working process?
7 responses

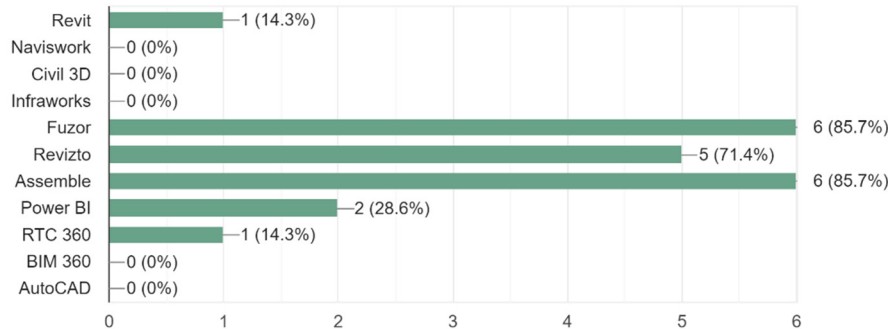

**Figure 3.** Summary of BIM and VDC technologies involved during working process.

In which project phase do you use it/them?
7 responses

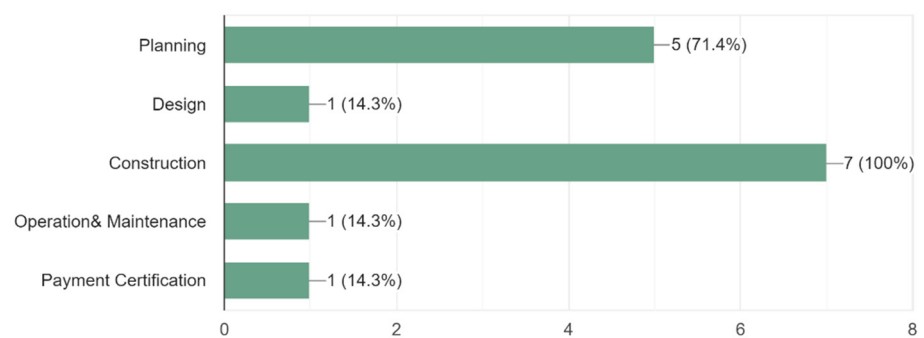

**Figure 4.** Summary of project phases when technologies were used.

In the pre-work-training-related questions section, 71.4% of participants thought that they often used the BIM/VDC technologies and 28.6% of participants thought that they used the BIM/VDC technologies very often. Although BIM and VDC technologies played such an important role in their daily work, 71.4% of participants had not received specific training before they took on their current positions. Based on this situation, 71.4% of participants thought that they needed more BIM and VDC training to become proficient in their current positions. These answers indicate that though BIM and VDC technologies were important in the construction field, training before work was insufficient to make practitioners more proficient and more highly efficient in reaching higher value. Therefore, specific training is necessary for employees to master these technologies. In this case, the educational side of the construction field could be encouraged to take responsibility for skills training by setting up specific courses, organizing internships, setting examinations, etc.

At the end of this research questionnaire, participants were asked what the benefits of BIM and VDC adoption in the AEC industry were. Most of the participants agreed that these technologies can obtain better insight into the project, improve communication and coordination, and reduce cost, time, and potential risks. Nevertheless, more than half of the responders hold the view that BIM and VDC technologies could not promise a more accurate estimation, a higher constructability, and a higher quality of projects and that the technologies did not provide significant assistance in these regards. In summary, accuracy, constructability, and quality can be seen as the goal in the further development of BIM and VDC.

## 5. Conclusions and Future Directions

This research studies the application of BIM and VDC technologies in a practical Atlantic Canada construction project through a case study. A one-on-one interview with a BIM/VDC

manager who oversaw the digital design in the case study building was designed. Additionally, a questionnaire for practitioners who participated in the case study building was designed. The two research methods aimed to obtain a general understanding of current BIM and VDC implementation from a general contractor's perspective in the Atlantic region of Canada. The interview questions were divided into three parts: technology, process, and culture. First, in the technology-related questions, BIM and VDC software were summarized and classified according to four factors: type, function, phase of utility, and owner/user (shown in Table 1). Second, in the process-related questions, the procedure of the application of BIM and VDC in projects was illustrated. There were several steps involved: initial investment, project planning, constructability analysis, establishing and modifying virtual BIM models where project stakeholders and clients were involved, project progress tracking, quality management, etc. Third, the culture-related questions were targeted towards the existing problems and challenges that barriered the advancement of BIM and VDC. Furthermore, based on previous questions, the responsibilities of industry, institutions, and government were also raised. In addition to the interviews, a research questionnaire focusing on BIM and VDC technology training was performed as a complement to this case study. The results of the questionnaire supported the technology-related and process-related information in the interview. It indicated that the specific skill training was insufficient and most of the practitioners thought they needed more training to be more professional.

Based on the research performed above, a new framework called the "Digital Construction Framework for the Future" is proposed to increase the adoption of BIM and VDC in today's Maritime AEC market and promote their further development. There are three individual parties involved in the framework: industry, institution and government, and education. These three parties play different roles in this framework but they do not function independently. Instead, they are bonded together closely; although the duties and functions are independent, every decision they make and every action they execute will affect others. They have a causal relationship, influence and supplement each other, forming a circular beneficial pattern of development (shown in Figure 5).

As the major division where BIM and VDC technologies are applied, the industry party should take the responsibility for ensuring that new technologies can be implemented in use throughout the whole project, involving the project team in the virtual environment. The more that BIM and VDC technologies are used in the AEC industry, the more benefits will be gained for both clients and project stakeholders, and more attention will be attracted to their advantages. Additionally, as shown in Table 2, one of the future challenges for BIM and VDC is increasing the estimation accuracy, constructability, and quality. To achieve this, some approaches could be applied: (1) BIM models can be linked with billing directly for better estimation and (2) the industry party should invest to mitigate the potential problems and risks during the design and construction stage. Well-planned strategy in the early stage can increase the constructability of the whole project, mitigate the hidden risks and conflicts, resulting in higher quality and profit. However, the industry party has limitations since it is only driven by project income. The institutions and government party in this framework are suggested to offer policy and financial help, driving the promotion of BIM and VDC technologies. Acting as the leader, it is important for them to create mandates and roadmaps as the guideline to lead the way forward. After that, institutions and government are suggested to follow the guideline, initiate the promotion of the BIM/VDC framework, and incentivize SMEs or large corporations to adopt technologies. With support from institutions and government, it will be easier to encourage the adoption of BIM/VDC technology and remove barriers to promotion. Last but not least, industry practitioners are the main division of the group driving the development of BIM and VDC technologies. According to the results from the research questionnaire, many employees hold the view that they lacked specific skill training, which they think is central for them to become more professional. The education party should be in charge of cultivating practitioners, ensuring that practitioners are equipped with sufficient expertise. It is proposed that the existing relative courses/degrees/programs should be revamped according to the current require-

ments. It is also vital to set up professional certificates in order to set up an evaluation standard. Thus, to summarize, these three parties need to function together to promote the development of BIM and VDC technologies. The education party teaches the related knowledge, offers competent employees for industry, institution, and government. At the same time, the industry party and the institution and government party could cooperate with the education party by offering internships and job opportunities for practical training. They could also give the education party suggestions on course modification according to the up-to-date industry requirements. Besides offering employment opportunities, the industry party is the implementer of institution and government programs, so it can give feedback to institutions and government and help them make better decisions about the development direction. Meanwhile, institutions and government give the industry and education parties policy and financial help, performing the main duty of encouraging people to apply these technologies. This is the digital construction framework for the future: all three of these parties (institutions and government, industry, and education) are inseparably interconnected and are supplementary to each other simultaneously, contributing to the progress of BIM and VDC technologies.

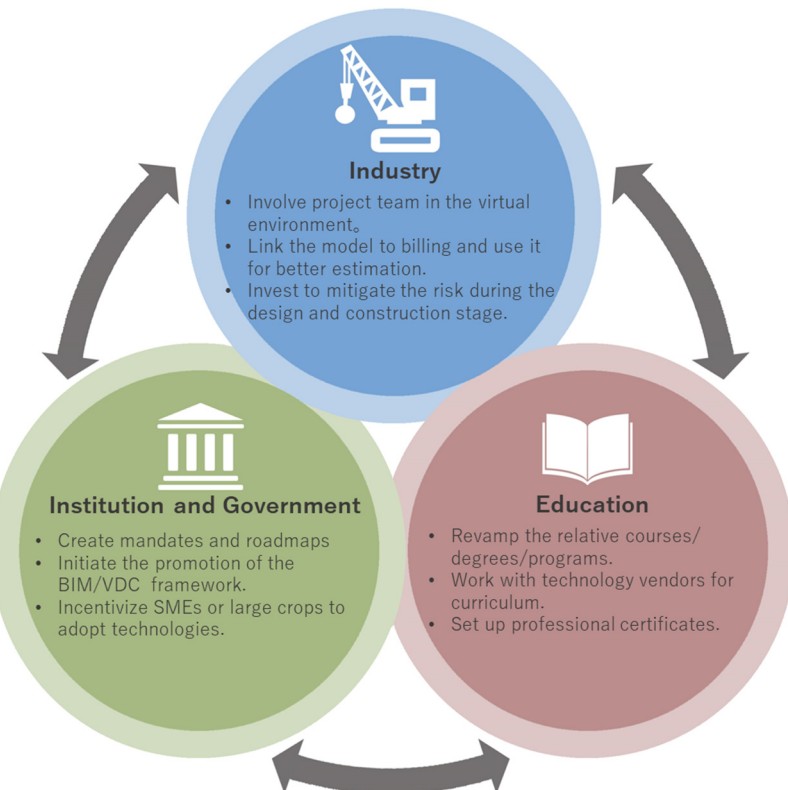

**Figure 5.** Digital Construction Framework for the Future.

**Table 2.** Benefits of BIM and VDC technologies.

| Benefits of BIM and VDC Technologies | No. | Percentage of Participants |
| --- | --- | --- |
| Improve communications and coordination | 6 | 85.7% |
| Improve estimation accuracy | 3 | 42.9% |
| Increase constructability | 3 | 42.9% |
| Risk mitigation | 5 | 71.4% |
| Save cost and resource | 5 | 71.4% |
| Improve efficiency and shorten project lifecycles | 5 | 71.4% |
| Reach higher quality results | 3 | 42.9% |
| Obtain better insight into the project | 7 | 100.0% |

It is important to note that this research (interview and survey) is a summary of the implementation of BIM and VDC technologies in real projects and how it is perceived by practitioners in Bird Construction. Additionally, from this, a framework is deduced to drive the adoption and development of these technologies. It could not represent the general views of all general contractors or companies due to the limitation of research group size and region restriction. However, as the studied company is one of the biggest and leading general contractors in construction in North America and the case project was constructed in Halifax, to a certain extent, it can be used to better the general understanding of the practical adoption of BIM and VDC in the Atlantic region of Canada. Future research can be conducted on the following topics: (1) surveys on a larger number of participants from wider fields (e.g., engineering, architecture design, material supply chain, transportation, and so on) to support and complement the conclusions of this paper and (2) additional projects could be researched as case studies in other areas of Canada or in other countries to summarize the general procedures of the adoption of BIM/VDC technology in real construction projects and to compare, modify, and complement the framework in this paper. This would increase the geographical coverage and decrease the limitation caused by location restrictions. Based on more research, a more generic and broadly applicable framework facing SMEs and large size companies could be designed to drive development forward.

**Author Contributions:** Conceptualization, M.A. and Z.L.; methodology, Z.L. and Z.C.; validation, Z.L.; formal analysis, M.A., Z.L. and Z.C.; investigation, M.A. and Z.L.; resources, M.A.; data curation, Z.L. and Z.C.; writing—original draft preparation, M.A., Z.L. and Z.C.; writing—review and editing, Z.L. and Z.C. All authors have read and agreed to the published version of the manuscript.

**Funding:** This research was funded by New Brunswick Innovation Foundation, grant number (RAI) RAI2021-0000000135 and The APC was funded by OSCO Research Chair Funding.

**Institutional Review Board Statement:** Not applicable.

**Informed Consent Statement:** Not applicable.

**Data Availability Statement:** Not applicable.

**Conflicts of Interest:** The authors declare no conflict of interest.

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
