# Peer review of "Integration of Building Information Modeling (BIM) and Virtual Design and Construction (VDC) with Stick-Built Construction to Implement Digital Construction: A Canadian General Contractor’s Perspective"

_buildings, doi:10.3390/buildings12091337_

Round 1

Reviewer 1 Report

The authors said, "It is important to note that this survey could not represent the general views of all practitioners in the AEC industry .......". This is correct and there shall be no "However" because it is a fact.

The study is only based on 1 case and f2f interviews with 7 different job holders of the case. 

The study is too simplistic to indicate that it is "A Canadian General Contractor's Perspective", needless to say, the general view of AEC industry on the use of Digital Construction. 

Figure 1 shows the Methodology clearly but the text does not tally with that. The authors should not relate the study to "all practitioners in the AEC industry" because there is insufficient evidence to justify that in a scientific paper. The study appears to be based on one project only and 7 participants of different job scopes make any statistical analysis from the questionnaire meaningless. It is difficult to convince readers that the study represents the company's view, not to mention, an indication of the AEC industry.

Author Response

Thank you for your time and feedback. The authors agree that this paper is just based on one single project with a general contractor. It should be defined as a case study and research rather than a general view of all Canadian general contractors or the AEC industry. The previous topic was general and unconvincing without specific data support.  However, the studied company is one of the largest general contractors in construction and property in Canada, and the case project was constructed by them in Halifax, to a certain extent, it can be used as a general situation and understanding of BIM and VDC practical adoption in the Atlantic region of Canada. Additionally, the research by its nature provides general guidance on BIM/VDC applications in the AEC industry through a lens of practitioners, so the results have both academic and practical values for those who plan to adopt the BIM/VDC in their organizations. However, more research with a larger number of practitioners is needed to support it. The authors changed the main topic into “the summary of the implementation of BIM/VDC technologies in one project, how it is perceived by practitioners in one of the leading general contractors in the Atlantic region of Canada", and then "proposed a framework to drive the application and development of the technology”. Also, the authors have carefully modified related content and added the research limitation and future work to explain it.

Line 88-95:

“Given the research is conducted based on a single general contractor in Canada, it does not have a reflection on the entire AEC industry. This is a limitation of this research. However, the research by its nature provides general guidance on BIM/VDC applications in the AEC industry through a lens of practitioners, so the results have both academic and practical values to those who plan to adopt the BIM/VDC in their organizations. To overcome this limitation, the authors will expand the survey areas in future work (e.g., covering other companies and different provinces in Canada).”

Line 579-597:

“It is important to note that this research (interview and survey) is a summary of the implementation of BIM and VDC technologies in real projects and how it is perceived by practitioners in Bird Construction. Additionally, from this, the framework is deduced to drive the adoption and development of these technologies. It could not represent the general views of all general contractors or companies due to the limitation of research group size and region restriction. However, the studied company is one of the biggest and leading general contractors in construction and property in Canada and the case project was constructed in Halifax, to a certain extent, it can be used as a general understanding of BIM and VDC practical adoption in the Atlantic region of Canada. Future research can be conducted on the following topics:(1) surveys for a larger number of participants from wider fields (e.g., engineering field, architecture design field, material supply chain, transportation, and so on) to support and complement the conclusions of this paper; (2) additional projects could be researched as case studies in other areas of Canada or in other countries to summarize the general procedures for the adoption of BIM/VDC technologies in real construction projects and to compare, modify and complement the framework of this paper. It can increase the geographical coverage and decrease the limitation caused by location restrictions. Based on more research, a more generic and broadly applicable framework facing small, middle or large size companies could be designed to drive the development forward.”

Reviewer 2 Report

Comment 1.

Please provide clear research gap and define research problem in the introduction – before outline of the paper (line 58). When defining this please move lines 297 – 299 to the introduction and clearly define research problem. Also, research problem needs to be define in the abstract – before line 13.

Comment 2.

I suggest moving lines 299-300 to the last paragraph of the introduction. This is outline of the paper which is final part of the introduction.

Comment 3.

It is not clear if survey participants were from the same company as person who was interviewed – please elaborate (line 116).

Comment 4.

It is not clear why importance of education was not included in interview but was studied using questionnaire – please elaborate (line 115 and 116).

Comment 5.

In Figure 1 is presented that literature review contribute to case study development, but currently it is not clear how and with what conclusions. Please provide table, figure, or bullets with main conclusions of the literature review at the end of the section 3. These conclusions need to be clearly connected with results and discussed in the conclusion.

Comment 6.

Please check data which you are providing in lines 202-206 (especially ‘634%’).

Comment 7.

I suggest listing all survey questions when explaining web-based questionnaire (the same as for interview questions). Currently it is not clear which is the number of survey questions (line 409 – before sentence ‘The first two questions…’).

Comment 8.

It is quite difficult to find the link between some survey questions and importance of education. Please define connection of listed survey questions with level of education and/or BIM/VDC knowledge and skills.

Comment 9.

Please insert Figure 5 after text where this Figure was first mentioned – after line 495.

Comment 10.

Please provide directions for future research and research limitations in section 5 (after line 537).

Comment 11.

Please correct some technical omissions in the text – e.g., space before reference – line 159; grey color – line 177, etc.

Author Response

Comment 1

Please provide clear research gap and define research problem in the introduction – before outline of the paper (line 58). When defining this please move lines 297 – 299 to the introduction and clearly define research problem. Also, research problem needs to be define in the abstract – before line 13.

Thank you for your time and feedback. The authors have carefully addressed this comment:

Line 13-15:

“However, currently, the promotion is relatively slow in North America. This paper focuses on developing an approach to drive the adoption of technologies through cooperation among project stakeholders and determining the way they will collaborate.”

Line 59-69:

“Many research cases showed the application of BIM/VDC technologies could bring not only more profit margins to the contractors but also non-financial advantages to society (e.g., non-renewable-resource-saving, pollution-reduction, and so on). Thus, the introduction, acceptance, and promotion of these new technologies should not be the responsibility of the industry alone, many other parties need to provide support. With more help from different fields, technology adoption can experience a faster and better development. Who should participate in promoting the use of these technologies and the way how they collaborate need to be researched. Therefore, the main topic of this paper is to clarify the responsibilities of each party and form a mutually supportive and cooperative relationship to drive the implementation and development of BIM and VDC technologies, through a case study.”

Comment 2

I suggest moving lines 299-300 to the last paragraph of the introduction. This is outline of the paper which is final part of the introduction.

Thank you for your time and feedback. The authors have carefully addressed this comment:

Line 85-88:

“At last, a new framework "Digital Construction Framework for Future" was proposed to conclude and describe the relationship and duties among these three parties, aiming at improving the adoption of digital technologies as the future direction.”

Comment 3

It is not clear if survey participants were from the same company as person who was interviewed – please elaborate (line 116).

Thank you for your time and feedback. The authors have carefully addressed this comment:

Line 444-447:

“In background-related questions, there were 7 participants. They came from Bird Construction and participated in the case study project in this questionnaire (one construction manager/project manager, one estimator, one senior construction superintendent, two project coordinators, and two site superintendents).”

Comment 4

It is not clear why importance of education was not included in interview but was studied using questionnaire – please elaborate (line 115 and 116).

Thank you for your time and feedback. The interview with the BIM/VDC manager was mainly aimed at understanding currently how the BIM/VDC technologies are used and function in real projects in a construction company because he was the person who oversaw the complete BIM/VDC application in the case study project. The education questions were put in the questionnaire part because it needs a larger size of practitioners to share their experiences to get the conclusion. However, the authors agree that the size of the questionnaire participants was small and the results obtained were not as good as expected. This result would have been more evident with a larger questionnaire volume. The future work we can conduct a larger size survey to optimize it.

Comment 5

In Figure 1 is presented that literature review contribute to case study development, but currently it is not clear how and with what conclusions. Please provide table, figure, or bullets with main conclusions of the literature review at the end of the section 3. These conclusions need to be clearly connected with results and discussed in the conclusion.

Thank you for your time and feedback. The authors have carefully thought about it. We have changed the Figure 1: the literature review did not contribute directly to the case study. It was used to get the definitions, development, advantages, and challenges of BIM and VDC technologies to understand the industry requirement, development history, and current state. With this background information, the framework could be better designed. Then the results from “BIM&VDC Literature Review” and “Case study” contributed together to obtain the framework. 

Comment 6

Please check data which you are providing in lines 202-206 (especially ‘634%’).

Thank you for your time and feedback. The authors have carefully checked the data. They are correct. However, about the number that you especially mention (Line 202-206, “The BIM ROI for different projects varied from 140% to 39,900%. On average, it was 1,633% for all projects and 634% for projects without a planning or value analysis phase.”), the original author explained that this large data spread was caused by peoples’ different calculation method in different projects. Thurs, he also said, “it is hard to conclude a specific range for BIM ROI”. The authors have carefully thought about this and decided to cite the average BIM ROI for projects instead. The authors have addressed this comment:

Line 221-222:

“The average BIM ROI for projects under the same study was 634%, which clearly depicts its potential economic benefits.”

Comment 7

I suggest listing all survey questions when explaining web-based questionnaire (the same as for interview questions). Currently it is not clear which is the number of survey questions (line 409 – before sentence ‘The first two questions…’).

Thank you for your time and feedback. The authors have carefully addressed this comment:

Line 424-442:

“The questionnaire was structured in four parts: (1) background: participants’ current position and field they were working in; (2) technology: software used in daily work; (3) pre-work training and (4) impacts. The detailed questions are listed as below:

Background-related questions:

  • What is your current position?
  • Which field are you mainly working in?

Technology-related questions:

  • What BIM and VDC technologies are involved during your working process?
  • In which project phase do you use it/them?
  • What’s the frequency you use it/them during your working process?

Pre-work-training-related questions:

  • Did you receive BIM and VDC training before you take on the current position? How long?
  • Do you feel you need more BIM and VDC training to become proficient in your cur-rent position?

Impact-related questions:

  • Compared with traditional project management, what do you think is the impact of digital construction management with BIM and VDC technologies?

In addition, the relative parts where the survey questions are mentioned have been changed according to the comment.”

Line 444-445:

“In background-related questions, there were 7 participants who came from the Bird Construction Company and participated in ….”

Line 450-451:

“In technology-related questions, the technologies mentioned by the VDC manager in the interview part were used as the options.”

Line 480:

“In the pre-work-training-related questions part……”

Comment 8

It is quite difficult to find the link between some survey questions and importance of education. Please define connection of listed survey questions with level of education and/or BIM/VDC knowledge and skills.

Thank you for your time and feedback. The authors wanted to use Line 483-486 (71.4% of participants had not received specific training before they take on their current positions. Based on this situation, 71.4% of participants thought they need more BIM and VDC training to become proficient in their current position.) to show the readers that if more training could be conducted, the practitioners can better utilize BIM/VDC technologies and bring more benefits to the industry, then promoting the application of the technologies. In addition, in this paper, we encourage the education party to take this responsibility. However. we didn’t link the training with education in the paper. The authors have carefully addressed this comment:

Line 486-491:

“These questions conducted that though BIM and VDC technologies were important in the construction field, training before work was insufficient to make practitioners more proficient, more high-efficient to reach a higher value. Therefore, specific training is necessary for employees to master these technologies. In this case, the educational side could be encouraged to take responsibility for skills training by setting up specific courses, organizing internships, setting examinations, etc.”

Comment 9

Please insert Figure 5 after text where this Figure was first mentioned – after line 495.

Thank you for your time and feedback. The authors have moved Figure 5 to the position you suggested.

Comment 10

Please provide directions for future research and research limitations in section 5 (after line 537).

Thank you for your time and feedback. The authors have carefully addressed this comment:

Line 88-95:

“Given the research is conducted based on a single general contractor in Canada, it does not have a reflection on the entire AEC industry. This is a limitation of this research. However, the research by its nature provides general guidance on BIM/VDC applications in the AEC industry through a lens of practitioners, so the results have both academic and practical values to those who plan to adopt the BIM/VDC in their organizations. To overcome this limitation, the authors will expand the survey areas in future work (e.g., covering other companies and different provinces in Canada).”

Line 579-597:

“It is important to note that this research (interview and survey) is a summary of the implementation of BIM and VDC technologies in real projects and how it is perceived by practitioners in Bird Construction. Additionally, from this, the framework is deduced to drive the adoption and development of these technologies. It could not represent the general views of all general contractors or companies due to the limitation of research group size and region restriction. However, the studied company is one of the biggest and leading general contractors in construction and property in Canada and the case project was constructed in Halifax, to a certain extent, it can be used as a general understanding of BIM and VDC practical adoption in the Atlantic region of Canada. Future research can be conducted on the following topics:(1) surveys for a larger number of participants from wider fields (e.g., engineering field, architecture design field, material supply chain, transportation, and so on) to support and complement the conclusions of this paper; (2) additional projects could be researched as case studies in other areas of Canada or in other countries to summarize the general procedures for the adoption of BIM/VDC technologies in real construction projects and to compare, modify and complement the framework of this paper. It can increase the geographical coverage and decrease the limitation caused by location restrictions. Based on more research, a more generic and broadly applicable framework facing small, middle or large size companies could be designed to drive the development forward.”

Comment 11

Please correct some technical omissions in the text – e.g., space before reference – line 159; grey color – line 177, etc.

Thank you for your time and feedback. The authors have carefully gone through the article and corrected the typos and omissions.

Reviewer 3 Report

The paper presents attitude construction compagnies facing BIM utilization in building process. The problem is well presented and methodology is fair adequate. References are strongly presented. The approach should have been gained to be more detailed. The two tools used ( interview and questionnaire) are very basic and look poor mathematically. However , despite the fact the issue of building construction process digitalization is very interesting, without better analysis, results look poor and specific for the fews geographical cases studies only. Therefore , it is difficult to generalise or extend results to small or middle size companies. The problem of the level of BIM knowledge and BIM comprehensive of each actor or stakeholder is also poorly adressed in the paper.

Author Response

Thank you for your time and positive feedback. The authors agree that this paper is limited in quantity (both the number of case study projects and the number of questionnaire participants). The small size of the questionnaire participants made the analysis not very convincing. However, the research by its nature provides general guidance on BIM/VDC applications in the AEC industry through a lens of practitioners, to a certain extent, the results have both academic and practical values to those who plan to adopt the BIM/VDC in their organizations. Additionally, we concluded them into our research limitations and future work. A larger-size survey can be conducted to complement/ modify the framework to make it more generic and broadly applicable. We have carefully addressed this comment:

Line 88-95:

“Given the research is conducted based on a single general contractor in Canada, it does not have a reflection on the entire AEC industry. This is a limitation of this research. However, the research by its nature provides general guidance on BIM/VDC applications in the AEC industry through a lens of practitioners, so the results have both academic and practical values to those who plan to adopt the BIM/VDC in their organizations. To overcome this limitation, the authors will expand the survey areas in future work (e.g., covering other companies and different provinces in Canada).”

Line 579-597:

“It is important to note that this research (interview and survey) is a summary of the implementation of BIM and VDC technologies in real projects and how it is perceived by practitioners in Bird Construction. Additionally, from this, the framework is deduced to drive the adoption and development of these technologies. It could not represent the general views of all general contractors or companies due to the limitation of research group size and region restriction. However, the studied company is one of the biggest and leading general contractors in construction and property in Canada and the case project was constructed in Halifax, to a certain extent, it can be used as a general understanding of BIM and VDC practical adoption in the Atlantic region of Canada. Future research can be conducted on the following topics:(1) surveys for a larger number of participants from wider fields (e.g., engineering field, architecture design field, material supply chain, transportation, and so on) to support and complement the conclusions of this paper; (2) additional projects could be researched as case studies in other areas of Canada or in other countries to summarize the general procedures for the adoption of BIM/VDC technologies in real construction projects and to compare, modify and complement the framework of this paper. It can increase the geographical coverage and decrease the limitation caused by location restrictions. Based on more research, a more generic and broadly applicable framework facing small, middle or large size companies could be designed to drive the development forward.”

Round 2

Reviewer 1 Report

Points well received.

Author Response

Thank you for your valuable feedback. We have again improved the manuscript and please see the re-submission as attached.

Reviewer 3 Report

The paper has been improved. And even if mathematical model is still absent, explanations given have been clearly done. Paper seems to be more mastered and explained. Recents references need to be added and cited.

Author Response

(The authors gave the same response as above.)
